# Preference for Comprehensive Dental Treatment under General Anesthesia among Parents with Previous Experience: A Cross-Sectional Study

**DOI:** 10.3390/children10111776

**Published:** 2023-11-02

**Authors:** Sara M. Bagher, Heba Jafar Sabbagh, Alaa Nadhreen, Najlaa M. Alamoudi, Abdullah Almushayt, Manal Al-Malik, Maha R. Al Shehri, Heba Mohamed Elkhodary

**Affiliations:** 1Pediatric Dentistry Department, Faculty of Dentistry, King Abdulaziz University, Jeddah 21589, Saudi Arabia; sbagher@kau.edu.sa (S.M.B.); anadhreen@kau.edu.sa (A.N.); nalamoudi@kau.edu.sa (N.M.A.); aamushayt@kau.edu.sa (A.A.); mraalshehri1@kau.edu.sa (M.R.A.S.); hkhodary@kau.edu.sa (H.M.E.); 2Pediatric Dental Department, King Fahad Armed Forces Hospital, Jeddah 21461, Saudi Arabia; manalmalik@kfafh.med.sa; 3Department of Pedodontics and Oral Health, Faculty of Dental Medicine for Girls, Al Azhar University, Cairo 11651, Egypt

**Keywords:** anesthesia, general, dental, parents, preference, behavior management technique

## Abstract

This study aims to assess whether parents of children who previously received comprehensive dental treatment under general anesthesia (GA) would prefer comprehensive dental treatment under GA over regular dental care if any of their other children required comprehensive dental treatment. In this cross-sectional study, parents of children who met the inclusion criteria were interviewed by a trained pediatric resident about parental-related factors as well as their satisfaction with their previous GA experience. Also, the factors related to the previously treated child were obtained from dental records. Statistical analysis was carried out, and the *p* value was set to 0.05. A total number of 306 parents were included. Although most parents, 293 (95.8%), showed satisfaction with the previous GA, 170 (58%) of the satisfied parents preferred regular care. Parents of children who were previously treated under GA for medical-related reasons (*p* = 0.018), fathers working in a governmental sector (*p* = 0.021), and families with low-average monthly income (*p* = 0.017) significantly preferred regular care. In conclusion, most parents were satisfied with the previous GA experience. Family income, fathers’ occupation, and medical-related factors can influence parental preference for comprehensive dental treatment under general anesthesia if any of their other children require comprehensive dental treatment.

## 1. Introduction

The main goal of pediatric dentistry is to provide an efficient, safe, and successful treatment for children and infants. This involves choosing an effective behavior management technique (BMT) to alleviate children’s dental anxiety and facilitate dental management [1]. The American Academy of Pediatric Dentistry (AAPD) offers various BMTs that can be divided into basic and advanced BMTs [2]. Basic BMTs involve communication guidance, tell-show-do, voice control, positive reinforcement, and distraction, while advanced BMTs comprise active and passive protective stabilizations, nitrous oxide sedation, oral sedation, and general anesthesia (GA) [2]. These behavior management techniques undergo a continuous reassessment process, which is mainly affected by the parental preference for the techniques used.

Nowadays, parental acceptance and preference for some BMTs, such as passive restraint and hand-over-mouth, has decreased, and most pediatric dentists have shifted toward pharmacological BMTs, including sedation and comprehensive dental treatment under GA [3,4,5,6,7,8].

Despite the risk and cost of comprehensive dental treatment under GA (including wait time that can delay and complicate treatment), comprehensive dental treatment under GA is effective as it enables the dentist to perform all the required dental treatments in one visit and decrease the possible psychological trauma associated with the need for long visits or multiple visits [9]. The fact that more treatments are covered in a single session while cutting down on treatment time emphasizes the idea of efficiency in the dental practice. This enhances the entire experience for the child while saving time for the dental staff and the parents [9].

Children with unique difficulties and special needs, who may find it difficult to participate during lengthy visits, benefit particularly from the all-encompassing approach under GA, making it the method of choice when the lack of cooperative ability is paired with the need for complex treatment [10].

In addition, comprehensive dental treatment under GA mainly focuses on restorative treatment followed by multiple follow-up appointments to emphasize good oral hygiene practices [11,12]. While attending the follow-up visits is highly recommended, the compliance was reported by many authors to be low and the prevalence of caries recurrence among children who received comprehensive treatment under GA is high. This prevalence is even higher among younger children and children with poor oral hygiene [11,12].

Therefore, a better understanding of the factors that affect parental acceptance and preference towards different BMTs among different populations with different cultural backgrounds is an important approach to promoting optimal treatment of children in pediatric dentistry [3,13,14].

Parental approval and informed consent are necessary before the selection and utilization of any BMTs for pediatric patients [15]. Few studies have investigated parental acceptance of different BMTs, and the factors associated with their acceptance [4,5,6,13,15,16,17]. These studies report that parents’ acceptance of particular BMTs varies based on the socio-economic status (SES) of the parent [8,15,16], cultural background [15], the urgency of the treatment [15,17,18], cost [15], level of the child’s cooperation [8,15], and parental previous experience [8,13]. However, these findings cannot be considered conclusive because the BMTs in these studies were either explained verbally to the parents or the parents were shown video tapes or photos of the techniques without any actual previous experience with such techniques.

The experience of comprehensive dental treatment under GA for one of the children can impact the parents’ inclinations, anticipations, and apprehensions regarding future dental treatments for their children. Also, they can provide unique viewpoints and outlooks on the process of decision making. Therefore, exploring these inclinations can provide more clarity on the different aspects of the parental decision-making process, as few previous studies investigated parental concerns regarding comprehensive dental treatment under GA and reported that most of the parents were concerned and were reluctant about comprehensive dental treatment under GA [19,20].

No previous study has discussed factors affecting parental preferences for comprehensive dental treatment under GA among parents with previous GA experience with one of their children. Therefore, the aim of this cross-sectional study was to assess whether parents of children who previously received comprehensive dental treatment under general anesthesia within the last four years would prefer GA instead of regular dental care in multiple visits if any of their other children required comprehensive dental treatment.

## 2. Materials and Methods

This cross-sectional study builds upon a previous project involving children who received dental treatment under general anesthesia (GA) between January 2016 and April 2017. The treatment was provided at two governmental centers in Jeddah: King Fahad Armed Forces Hospital (KFAFH) and King Abdulaziz University Dental Hospital (KAUDH) [15]. These centers, which are funded by the Ministry of Health, offered comprehensive treatment free of charge to all patients. The approval from the ethics committees were granted from the Local Research and Ethical Committee at KFAFH (# 001-37925; approved in October 2013) and the Research Ethics Committee of the Faculty of Dentistry at King Abdulaziz University (REC No. 062-15; approved in May 2015). These committees reviewed the study protocol and determined that it met the necessary ethical standards for conducting research involving human participants prior to the data collection phase of the study.

All children who received comprehensive dental treatment under GA in the previously mentioned centers were recommended to attend post-treatment follow-up visits. The follow-up visits were scheduled after one week and then at 1-, 3-, 6-, 12-, 18-, and 24-month intervals. The children received clinical and radiographic evaluations of the treatment provided, oral hygiene reinforcement, and, in some cases, dental treatment for new lesions or failed treatments. Children who attended at least four of the recommended follow-up visits throughout the two-year period were considered “compliant”. Those who attended less than four follow-up visits were considered “non-compliant”.

The hospital records of the children who received comprehensive dental treatment under GA at KAUDH or KFAFH between January 2016 and April 2017 were reviewed retrospectively by a trained pediatric resident in September 2021. Parents were contacted, and the aim of the study was introduced and explained to them over the phone. For those who agreed to participate, an interview appointment was scheduled, and an Arabic consent form was obtained from the parents before participation. The scheduled interview appointment was performed by a single trained pediatric resident by utilizing a previously validated questionnaire [16]. Parent-related factors including parents’ age, level of education, occupation, family average monthly income, and parental satisfaction with the previous comprehensive dental treatment under GA for their child were obtained. The family’s average monthly income was based on the central statistics and information provided on the website of Saudi Arabia for family incomes, where <7000 Saudi Riyals (SR) is considered low income; 7000–10,000 SR is low-to-middle income; 10,000–16,000 SR is middle-to-high income; and >16,000 SR is high income [16]. Parental education was divided into three groups: illiterate, those with high-school-level education or less, and those with university and higher degrees. Parents were also asked if any of their other children required comprehensive dental treatment and if they would prefer the treatment to be provided under GA or by regular dental care distributed throughout multiple visits.

The factors related to the previously treated child were obtained from the dental records, including the age of the child at which her/she received the comprehensive dental treatment under GA (<six years and ≥six years), the reason for seeking comprehensive dental treatment under GA, the hospital in which the previously treated child received his GA dental treatment, the average GA appointment waiting time, the type of hospital entry (daycare or admission), and the frequency of attending the recommended follow-up visits after dental treatment under GA.

### Statistical Analysis

Statistical analysis was carried out using the Statistical Package for Social Science program for Windows version 20 (SPSS, Chicago, IL, USA). A chi-square test was performed to detect the significance of the different variables on parental preference for comprehensive dental treatment under GA. Binary logistic regression, reported as odds ratios (ORs) and 95% confidence intervals (CIs), was used to estimate covariate-adjusted associations with the dependent variable (parental acceptance of GA). The covariates included child age, the key sociodemographic predictors (family income), GA waiting time and the reason for treating the child’s teeth under GA. The *p*-value cut for significance was set to 0.05.

## 3. Results

A total of 634 children had received comprehensive dental treatment under GA between January 2016 and April 2017 in both centers. Upon contacting them, 259 (40.9%) did not answer and 69 (10.9%) parents refused to participate in the study. Therefore, a total of 306 parents participated in the study, with a 48.3% response rate. Of the participating parents, 137 (44.8%) belonged to the low- to middle-average monthly income category, and 129 (42.2%) had five or fewer children. The mean age of children who previously received comprehensive dental treatment under GA at the time of the comprehensive dental treatment under GA was 5.25 ± 1.73 years, and at the interview appointment was 8 ± 2.15 years. The results were reported following the Strengthening the Reporting of Observational studies in Epidemiology (STROBE) guidelines [21].

The results showed that 170 (55.6%) of parents preferred regular dental care in multiple visits if needed for any of their other children. Fathers working in the governmental sector (*p* = 0.021) and families with low-average monthly income (*p* = 0.017) showed a statistically significant preference for regular dental care in multiple visits. Most of the parents (293 (95.8%)) were satisfied with the previous comprehensive dental treatment under GA. However, further analysis showed that 160 (54.6%) of the 293 satisfied parents preferred regular dental care in multiple visits if needed again for any of their other children, but the difference was not statistically significant (*p* = 0.11). The parental treatment preference distribution according to parental-related factors and previous experience with comprehensive dental treatment under GA is presented in Table 1.

Significantly, parents of the children who previously received comprehensive dental treatment under GA due to medical-related problems (*p* = 0.018) were more likely to prefer regular dental care in multiple visits if comprehensive treatment is needed again. The parental treatment preference distribution according to the previously treated child-related factors is presented in Table 2.

Although not statistically significant, the results also indicated that parents with an unsatisfactory previous experience with comprehensive dental treatment under GA and parents of children requiring hospitalization tended to prefer regular dental care in multiple visits over comprehensive dental treatment under GA if any of their other children required comprehensive dental treatment.

Binary logistic regression was conducted to adjust the associations of covariance with the parental preference for GA. Low-average monthly income showed significantly decreased parental preference for GA compared to regular dental care in multiple visits (*p* = 0.024; AOR: 0.381). On the other hand, parents of children treated under GA for behavior reasons statistically significantly preferred GA compared to parents of medically compromised children. (*p* = 0.019; AOR: 2.308) (Table 3).

## 4. Discussion

Most of the parents with previous experience in comprehensive dental treatment under GA with one of their children were satisfied with the experience, but more than half of them preferred regular dental care in multiple visits if comprehensive dental treatments were needed by any of their other children. Four factors related to that decision showed a statistically significant relationship to parental preference: the medical condition of the previously treated child, the reason for the comprehensive dental treatment under GA, the fathers’ occupation, and the family’s average monthly income.

Multiple previous studies evaluated parental satisfaction with the comprehensive dental treatment provided to their children [18,22,23,24], and a high level of satisfaction (over 95%) was reported among the studies. The results of these studies were consistent with the current study results, in which 95% of the parents were satisfied with their child’s GA experience. The study also reported that 55.6% of the parents preferred regular dental care in multiple visits if any of their other children required comprehensive dental treatment after they had previous experience. However, the results are inconsistent with a previously published study by El Batawi et al., 2014 [24], who evaluated Saudi parents’ satisfaction after comprehensive dental rehabilitation under GA and reported that most of the parents 97.44% preferred dental treatment under GA if it was needed again and only 2.55% reported that they would not repeat the GA experience and would prefer multiple visits under sedation instead. This difference can be due to the fact that the children in the current study were from different educational and socio-economic levels and were treated in governmental hospitals for free, while in the study conducted by El Batawi et al., 2014, [24] they were treated in private hospitals which were typically associated with families with average or high monthly income.

The current study reports that parents of children who received comprehensive dental treatment under GA due to behavioral problems tended to prefer dental treatment under GA, but parents of children who received it due to medical problems preferred regular dental care in multiple visits. This agreed with a previous study published by Castro et al., 2016 [18] to assess parental acceptance of different BMTs among parents of medically compromised children and reported that all the participating parents considered communicative management totally acceptable while dental treatment under GA was accepted by 75.90% of the parents. That could be attributed to the parent’s concern about the child’s well-being and fear of any adverse medical events that might occur due to their pre-existing medical condition.

In addition, the age of the child was found to be related to GA preference as a BMT. As reported in a study conducted by Popova et al. in 2023 [25], dental treatment under GA was more acceptable to the parents if the child was older than 7 years of age. On the contrary, another study reported that parents of very young children (2–3 years old) or children with no previous dental experience would prefer pharmacological techniques, while parents of older children would place greater dependence on the child’s ability to cope and highly prefer nonpharmacological methods [8]. Therefore, it is recommended to include the child’s age in future research when assessing parental acceptance.

Therefore, in future studies that evaluate different confounding factors affecting parental acceptance and preference for different BMTs of the parents of medically compromised children, the investigators are recommended to have a better understanding of the factors that potentially play a role in the parental acceptance and preference in their situation. Parental reluctance to repeat the treatment experience under GA might be attributed to multiple reasons, such as fearing for the child’s safety, deeming another experience as unnecessary, or the lack of communication with the dentist on its necessity. Further investigation should be performed to enhance the informed decision-making process by addressing the parents’ possible concerns as they attempt to make the decision. Also, once the parents are satisfied with the decision they made, acceptance and compliance with the treatment and the maintenance thereafter might also be enhanced.

In the current study, parents with lower average monthly income levels preferred regular dental care throughout multiple visits, while parents of children with high average monthly income preferred dental treatment under GA. This disagrees with a study published by Patel et al., 2016 [9], which reported that as the cost of the treatment increased, parental acceptance of such treatment decreased. In the current study, both included hospitals are governmental, and the treatment provided was completely free, which could justify the parent preferences as reported. In addition, a similar result was reported by a previous study conducted in Saudi Arabia on nitrous oxide (N_2_O) sedation. The study reported that parents of low family income tend to prefer the use of papoose and/or papoose with N_2_O compared to the use of N_2_O alone [26]. Also, parental socio-economic status and parental style were found to affect their preference for the different behavior techniques used with their children [26].

In the current study, parents who were compliant and attended the recommended follow-up visits tended to favor regular dental care in multiple visits. These findings reflect the necessity of post-treatment follow-up visits, during which children and parents will get oral hygiene reinforcement and anticipatory counseling and become more aware of the influence of dental health on their children’s health and quality of life. As a result, they may feel more obligated to provide their children with excellent dental care and to offer the necessary treatment in a less invasive manner once one of their children requires comprehensive treatment. In addition, the child will get more accustomed to the dental visits and get more familiar with the dentist, which will ease the treatment process and significantly decrease dental-related anxiety [27], which will lead to a lesser need for GA as the form of behavior management. De Menezes et al. concluded that when children were gradually introduced to the dental environment through a series of sequential dental visits of varying types, a reduction in their levels of dental anxiety over a 14.5-month period was noticed [27].

The study conducted by Delikan et al., 2019, [23] to assess parental satisfaction after dental treatment under GA reported a significant correlation between waiting time before the comprehensive dental treatment under GA and the parent’s satisfaction with the experience. In this study, most of the children had to wait between three to six months for the GA appointment, and although the duration of waiting time did not significantly affect parental preference, those who had to wait for a shorter duration had more tendency to prefer regular dental care in multiple visits. In a study conducted by Okuji et al. [28], they concluded that during the period of waiting for comprehensive dental treatment under GA, the oral health of pediatric patients showed a continuous decline. Over the course of the three-month waiting period, these young patients experienced either the onset of new symptoms or the exacerbation of existing ones, resulting in the need for more extensive dental care under GA compared to the initial treatment plan. This highlights the significant impact of the waiting duration on the number of visits prior to comprehensive dental treatment under GA [28].

One of the limitations of this study was the relatively low response rate. Another limitation was that the study did not evaluate and compare parental preference before and after their experience with comprehensive dental treatment under GA; thus, a cohort study is recommended to assess changes in parental preference. Also, the reason behind the unsatisfied previous experience was not investigated. In addition, the child’s age at the time of comprehensive dental treatment under GA and scheduled interview appointment should be included as one of the confounders that affect parental preference for GA. Given these limitations, further future studies evaluating parental preference for dental treatment under GA and assessing the factors associated with their acceptance in specific populations are recommended to eliminate the confounding variables that might affect their preferences.

## 5. Conclusions

Although most of the parents were satisfied with the previous GA experience, only half of them preferred regular dental care if needed again for any of their other children. Factors that had a statistically significant relationship to parental preference were the medical condition of the child, the reason behind the comprehensive dental treatment under GA, the father’s occupation, and the family’s average monthly income. Regular dental care was preferred by families with low-average monthly income, fathers working in the government sector and parents of children who were previously treated under GA for medical-related reasons.

## Figures and Tables

**Table 1 children-10-01776-t001:** Parental treatment preference distribution according to parental-related factors and previous experience with comprehensive dental treatment under general anesthesia (N = 306).

Variables	Parental Preference		*p*-Value
	Comprehensive Dental Treatment under GA N (%)	Regular Dental Care in Multiple Visits N (%)	
**Maternal age**			
<31 years	26 (42.6)	35 (57.4)	0.48
31–39	70 (47.4)	76 (52.1)	
≥40 years	40 (40.4)	59 (59.6)	
**Paternal age**			
<40	56 (48.3)	60 (51.7)	0.6
41–50	56 (42.4)	76 (57.6)	
≥51	24 (41.4)	34 (58.6)	
**Child age at the GA**			
<6	98 (43.2)	129 (56.8)	0.45
≥6	38 (48.1)	41 (51.9)	
**Maternal education**			
University and higher degree	48 (44.9)	59 (55.1)	0.45
High school	74 (42.5)	100 (57.5)	
Illiterate	14 (56)	11 (44.0)	
**Paternal education**			
University and higher degree	29 (42.6)	39 (57.4)	0.94
High school	103 (45)	126 (55.0)	
Illiterate	4 (44.4)	5 (55.6)	
**Maternal occupation**			
Housewife	103 (43.5)	134 (56.5)	0.79
Private sector	4 (44.4)	5 (55.6)	
Government sector	29 (48.3)	31 (51.7)	
**Paternal occupation**			
Military	97 (51.0) *	93 (48.9)	0.021 *
Private sector	5 (27.8)	13 (72.2)	
Government sector	11 (26.8)	30 (73.2)	
Retired	23 (40.4)	34 (59.6)	
**Family average monthly income**			
Low	21 (32.8)	43 (67.2)	0.017 *
Low to middle	60 (43.8)	77 (56.2)	
Middle to high	33 (51.6)	31 (48.4)	
High	22 (53.7)	19 (46.3)	
**Previous GA experience**			
Unsatisfactory	3 (23.1)	10 (76.9)	0.11
Satisfactory	133 (45.4)	160 (54.6)	

* Statistically significant using chi-square at *p* ≤ 0.05. GA: General anesthesia.

**Table 2 children-10-01776-t002:** Parental treatment preference distribution according to the previously treated child-related factors (N = 306).

Variables	Parental Preference		*p*-Value
	Comprehensive Dental Treatment under GA N (%)	Regular Dental Care in Multiple Visits N (%)	
**Reasons for comprehensive dental treatment under GA**			
Rampant caries	48 (44.9.5)	59 (55.1)	0.018 *
Behavioral problems	71 (50.7)	69 (49.3)	
Medical problems	17 (28.8)	42 (71.2)	
**Compliant with attending follow-up visits**			
No	28 (54.9)	23 (45.1)	*p* = 0.1
Yes	108 (42.4)	147 (57.6)	
**Comprehensive dental treatment under GA, waiting time in months**			
0–2	37 (39.8)	56 (60.2)	0.43
3–6	39 (39.8)	56 (60.2)	
7–12	40 (51.3)	38 (48.7)	
>12	17 (53.1)	15 (46.9)	
**Hospital Admission**			
Hospitalization	13 (31.0)	29 (69.0)	0.058
Daycare	123 (46.6)	141 (53.4)	

* Statistically significant using chi-square at *p* ≤ 0.05. GA: General Anesthesia.

**Table 3 children-10-01776-t003:** Binary logistic regression for the association between parental preference for comprehensive dental treatment under general anesthesia (dependent factor) and child age at GA, family average monthly income, the reason for comprehensive dental treatment under general anesthesia, and general anesthesia waiting time.

Variables		*p* Value	AOR	95% CI	
Child age at the GA	<6	0.293	0.729	0.405	1.313
	≥6		1		
Family average monthly income	Low	0.024 *	0.381	0.165	0.990
	Low to middle	0.2307	0.629	0.306	1.325
	Middle to high	0.595	0.804	0.360	1.955
	High		1		
Reasons for comprehensive dental treatment under GA	Rampant caries	0.085	1.963	0.912	4.229
	Behavioral problems	0.019 *	2.308	1.149	4.636
	Medical problems				
GA waiting time in months	0 to 6	0.141	0.689	0.419	1.131
	>6		1		

* Statistically significant using chi-square at *p* ≤ 0.05. GA: General Anesthesia.

## Data Availability

The data used in this study is available upon reasonable request from the corresponding author.

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
