# Peer review of "Preference for Comprehensive Dental Treatment under General Anesthesia among Parents with Previous Experience: A Cross-Sectional Study"

_children, 2023, doi:10.3390/children10111776_

Round 1
Reviewer 1 Report (Previous Reviewer 1)
Comments and Suggestions for Authors
Review report
Parental preference of comprehensive dental treatment under general anaesthesia among parents with previous experience: A cross sectional study”.
Points to be addressed:
Affiliations: use the same number if the affiliation is the same. In this paper, the affiliations are 3, therefore 3 numbers are required. Write the affiliation once and then all the mail of the Authors. Please, look at the Instructions for Authors.
Abstract
· Please, remove italic from the abstract.
· Use p value instead of P-value.
· A maximum of 200 words are required for the abstract, please synthetize the introduction.
· Lines 40-42: there is a repetition of the results in the abstract.
· Which are the conclusions? It should be added that the factors influencing the choice could be economic, medical, and so on.
Introduction
- It should be also added that in general patients needing GA are at high risk of developing other caries and oral hygiene education is very difficult.
Materials and methods
· Please, follow STROBE guidelines for observational study for the writing of the manuscript (https://www.strobe-statement.org)
· Approval number of Local Research and ethical Committee at KFAFH?
· I do not like “regular” and “irregular” definitions.
· Please, remove 2.1 heading, as it is the only one in the M&Ms section, or provide more headings.
Discussion
· How about considering comparisons with nitrous oxide and hypnotherapy? (DOI: 10.23804/ejpd.2023.1932).
- Recent references should be added (DOI: 10.22514/jocpd.2023.034; DOI: 10.22514/jocpd.2022.030).
· Line 237: the age of the child was not included in the data.
Author contributions: authors should be abbreviated with initials.
Institutional Review Board statement: the approval number should be provided.
Thank you for the effort.
Author Response
Please see the attachment.

Reviewer 2 Report (Previous Reviewer 2)
Comments and Suggestions for Authors
Dear Authors;
As you alluded to in your comments, concerns regarding timeline inconsistency are evident, and it is essential to ensure that ethical approval is consistent with the specific period of data acquisition. In your investigation, you mention two decisions made by a two different ethics committee in 2013 and 2015. Why did you consider the need to take a second ethics committee in addition to the one you took in 2013? My opinion is that this retrospectively planned research should be conducted with a current ethics committee decision and that a single, updated ethics committee decision should cover all the study's specifics.
In addition, between lines 161 and 173, there are still inconsistent study results.
Author Response
Please see the attachment.

Reviewer 3 Report (New Reviewer)
Comments and Suggestions for Authors
This manuscript assessed whether parents of children who previously received comprehensive dental treatment under general anaesthesia would prefer comprehensive dental treatment under general anaesthesia instead of regular dental care in multiple visits if any of their other children required comprehensive dental treatment. Moreover, factors associated with parental acceptance and preference were analysed. Three hundred and six parents participated in the study. The manuscript contains four keywords, two tables, and twenty-five references. Overall, it is a correct paper.
General comments
This study highlights the use of general anaesthesia for one-session dental treatment in children as a choice to regular dental care in multiple visits. The study had several weaknesses, highlighted by the own authors as limitations (relatively low response rate; parental preferences before and after their experience with comprehensive dental treatment under general anaesthesia could not be evaluated). The study has been approved by two different Ethics Committees.
The study is methodologically well-established. The data management is basic but appropriate according to the approach of the study. The results are well presented, being easy to read and interpret them. In the discussion section, the results of this study are adequately contrasted with those obtained by other researchers. A justifying explanation of the results is also provided. The manuscript also includes an appropriate conclusions section.
Some further remarks are made on different sections of the manuscript.
Keywords
The manuscript presents four keywords. For keywords, where possible, please use Medical Subject Headings terms (MeSH Terms). Strictly, none of them is a MeSH term. An alternative MeSH term proposed could be “anesthesia, general” better than “general anaesthesia”. Nevertheless, these suggestions about keywords are optional, not mandatory.
Other manuscript sections
Page 3, line 155. In the statistical analysis section, please indicate the version of the SPSS statistical software used.
To set statistical significance, the p-value was equal to 0.05 (page 4, line 159) or less than 0.05 (table 1)? Please, clarify this.
To make text understanding easier, if the author's name appears in the text, place the reference number immediately after the name, not at the end of the sentence or paragraph.
References
Total number of the manuscript references: 25.
Great! The reference format matches the journal’s reference format (ACS style guide).
Tables
Total number of the manuscript tables: 2.
The tables have appropriate titles and information. Please, consider explaining the abbreviation “GA” in the two tables.
In Table 1, why are different age ranges used for mothers and fathers?
Round 2
Reviewer 1 Report (Previous Reviewer 1)
Comments and Suggestions for Authors
Dear Authors,
Thank you for providing the revised version.
Paragraph 2.1 is still present, after removing it the article could be published.
Thank you for your commitment.
Reviewer 2 Report (Previous Reviewer 2)
Comments and Suggestions for Authors
I am still thinking the ethics committee decision should be more current than the date of the retrospective study.
My opinion is that this retrospectively planned research should be conducted with a current ethics committee decision and that a single, updated ethics committee decision should cover all the study's specifics.
This manuscript is a resubmission of an earlier submission. The following is a list of the peer review reports and author responses from that submission.
Round 1
Reviewer 1 Report
Comments and Suggestions for Authors
Dear Authors,
I have been invited to review your work entitled “Parental preference of comprehensive dental treatment under general anaesthesia among parents with previous experience: A cross sectional study”. The study was well conducted, however there are some issues that deserve revisions for the acceptance of the manuscript. Please, provide a point-by-point response, highlighting the corrections with a different color mark for each reviewer.
Affiliations: please, provide only the affiliations and not also the official title of the Authors.
Abstract
· The abstract should be structured without headings.
· You should add that data underwent statistical analysis with significance threshold.
Introduction
- Novel literature with interesting data can be added (DOI: 10.22514/jocpd.2023.034; DOI: 10.22514/jocpd.2022.030).
- It should be also added that those kinds of patients are at high risk of developing other caries and oral hygiene education is very difficult (DOI: 10.22514/jocpd.2023.024)
- Lines 93-104: these lines should be deleted, as you have concluded the Introduction section stating the objective of the study.
Materials and methods
· How about parents’ education? Only four categories?
Discussion
· The discussion should be reduced as it is very long and the readers can get confused.
Thank you for the effort.
Comments on the Quality of English Language
English revision is required before publication.
Reviewer 2 Report
Comments and Suggestions for Authors
To the author:
Nowadays, researchers have focused on the parental acceptance of advanced behavior guidance techniques used in pediatric dentistry and the reasons for rejection. In your study, you intended to determine if parents with prior experience with general anesthesia would accept this procedure for their other children. Congratulations on conducting research on this timely and special subject. However, as a consequence of my review of your manuscript, I would like to make the following recommendations:
• Initially, you stated that the date of your ethics committee's approval was 2015. Nevertheless, you stated that the study's retrospective data analysis occurred between 2016 and 2017. In such a circumstance, I believe it would be preferable to obtain an updated ethical approval.
• The manuscript contains numerous typographical and grammatical errors. I recommend that a native English speaker read your manuscript. This will make some of your statements easier to comprehend. (See, for instance, your statement on lines 192 and 199.).
· In the Discussion section, you stated that those who routinely attend the post-treatment follow-up visits recommended in your manuscript prefer 'routine dental care at multiple visits'. However, according to your study's findings, there is no statistically significant difference between the general anesthesia and multiple visit treatment options for patients who were routinely followed up (p=0.1).
· • In the Discussion section, you mentioned references indicating that the age of the child influences the decision of the parents to accept to general anesthesia. However, you did not provide any information regarding the ages of the operated children and their siblings in the manuscript. It may be appropriate to include this circumstance in the study's limitations.
· How did you classify the number of children each parent has? Why did you specify a range of less than five? Isn't there a distinction between having two and five children? Please elaborate.
· What does the * in the military line mean in the paternal occupation section in Table 1? Shouldn't you put the * to the government line ?
· • The expression number and percentage of patients (n=170, 55.6%) in line 158 is incorrect according to Table 1. Please write the correct number and percentage.
· • It may not be appropriate to write the following sentence in the conclusion part. ‘’ Most of the parents were satisfied with the previous GA experience but preferred regular dental care if needed again for any of their other children.’’. Because according to your data, there is no statistically significant difference between multiple visit treatment or general anesthesia preferences of patients who are satisfied with their GA experience (p=0.11).
· Please include your results in more detail in the discussion section. How might the paternal employment in a government institution influence this decision? You stated that the treatments in the hospitals where the study was conducted are free of charge. What could be the reasons for low income level to prefer multiple visit treatment? What might the parents of patients who had previously been operated for medical reasons have encountered during and after the procedure? What factors might have influenced this decision?
Comments on the Quality of English Language
The manuscript contains numerous typographical and grammatical errors. I recommend that a native English speaker read your manuscript. This will make some of your statements easier to comprehend. (See, for instance, your statement on lines 192 and 199.).
Reviewer 3 Report
Comments and Suggestions for Authors
I congratulate the authors of the present study for their time and efforts. Unfortunately, the study has a massive problem regarding study design and ethics. The study has a mixed design: retrospective& prospective. The problem is that a researcher can do a retrospective study on the data before the ethics approval date, not after. However, the retrospective area is without ethical approval. This isn't very clear and raises questions about the method and ethical responsibilities.
Comments on the Quality of English Language
The quality of English is moderate. Some sentences are long and hard to understand. Those need to be rewritten.